# L4: Practical loss-based stepsize adaptation for deep learning

**Michal Rolínek and Georg Martius**
Max-Planck-Institute for Intelligent Systems Tübingen, Germany
`michal.rolinek@tuebingen.mpg.de` and `georg.martius@tuebingen.mpg.de`

## Abstract

We propose a stepsize adaptation scheme for stochastic gradient descent. It operates directly with the loss function and rescales the gradient in order to make fixed predicted progress on the loss. We demonstrate its capabilities by conclusively improving the performance of Adam and Momentum optimizers. The enhanced optimizers with default hyperparameters consistently outperform their constant stepsize counterparts, even the best ones, without a measurable increase in computational cost. The performance is validated on multiple architectures including dense nets, CNNs, ResNets, and the recurrent Differential Neural Computer on classical datasets MNIST, fashion MNIST, CIFAR10 and others.

## 1 Introduction

Stochastic gradient methods are the driving force behind the recent boom of deep learning. As a result, the demand for practical efficiency as well as for theoretical understanding has never been stronger. Naturally, this has inspired a lot of research and has given rise to new and currently very popular optimization methods such as Adam [9], AdaGrad [5], or RMSProp [22], which serve as competitive alternatives to classical stochastic gradient descent (SGD).

However, the current situation still causes huge overhead in implementations. In order to extract the best performance, one is expected to choose the right optimizer, finely tune its hyperparameters (sometimes multiple), often also to handcraft a specific stepsize adaptation scheme, and finally combine this with a suitable regularization strategy. All of this, mostly based on intuition and experience.

If we put aside the regularization aspects, the holy grail for resolving the optimization issues would be a widely applicable automatic stepsize adaptation for stochastic gradients. This idea has been floating in the community for years and different strategies were proposed. One line of work casts the learning rate as another parameter one can train with a gradient descent (see [2], also for a survey). Another approach is to make use of (an approximation of) second order information (see [3] and [19] as examples). Also, an interesting Bayesian approach for probabilistic line search has been proposed in [13]. Finally, another related research branch is based on the "Learning to learn" paradigm [1] (possibly using reinforcement learning such as in [12]).

Although some of the mentioned papers claim to "effectively remove the need for learning rate tuning", this has not been observed in practice. Whether this is due to conservativism on the implementor's side or due to lack of solid experimental evidence, we leave aside. In any case, we also take the challenge.

Our strategy is performance oriented. Admittedly, this also means, that while our stepsize adaptation scheme makes sense intuitively (and is related to sound methods), we do not provide or claim any theoretical guarantees. Instead, we focus on strong reproducible performance against optimized

baselines across multiple different architectures, on a minimum need for tuning, and on releasing a prototype implementation that is easy to use in practice.

Our adaptation method is called **L**inearized **L**oss-based optima**L** **L**earning-rate (L$^4$) and it has two main features. First, it operates directly with the (currently observed) value of the loss. This eventually allows for almost independent stepsize computation of consecutive updates and consequently enables very rapid learning rate changes. Second, we separate the two roles a gradient vector typically has. It provides both a local linear approximation as well as an actual vector of the update step. We allow using a different gradient method for each of the two tasks.

The scheme itself is a meta-algorithm and can be combined with any stochastic gradient method. We report our results for the L$^4$ adaptation of Adam and Momentum SGD.

## 2 Method

### 2.1 Motivation

The stochasticity poses a severe challenge for stepsize adaptation methods. Any changes in the learning rate based on one or a few noisy loss estimates are likely to be inaccurate. In a setting, where any overestimation of the learning rate can be very punishing, this leaves little maneuvering space.

The approach we take is different. We do not maintain any running value of the stepsize. Instead, at every iteration, we compute it anew with the intention to make maximum possible progress on the (linearized) loss. This is inspired by the classical iterative Newton's method for finding roots of one-dimensional functions. At every step, this method computes the linearization of the function at the current point and proceeds to the root of this linearization. We use analogous updates to locate the root (minimum) of the loss function.

The idea of using linear approximation for line search is, of course, not novel, as witnessed for example by the Armijo-Wolfe line search [15]. Also, and more notably, our motivation is identical to the one of the Polyak's update rule [16], where the loss-linearization (Eq. 2) is already proposed in a deterministic setting as well as the idea of approximating the minimum loss.

Therefore, our scheme should be thought of as an adaptation of these classical methods for the practical needs of deep learning. Also, the ideological proximity to provably correct methods is reassuring.

### 2.2 Algorithm

In the following section, we describe how the stepsize is chosen for a gradient update proposed by an underlying optimizer (e. g. SGD, Adam, momentum SGD). We begin with a simplified core version.

Let $L(\theta)$ be the loss function (on current batch) depending on the parameters $\theta$ and let $v$ be the update step provided by some standard optimizer, e. g. in case of SGD this would be $\nabla_\theta L$. Throughout the paper, the loss $L$ will be considered to be non-negative.

For now, let us assume the minimum attainable loss is $L^{\min}$ (see Suppl. Section 2.4 for details). We consider the stepsize $\eta$ needed to reach $L^{\min}$ (under idealized assumptions) by satisfying

$$L(\theta - \eta v) \stackrel{!}{=} L^{\min}. \tag{1}$$

We linearize $L$ (around $\theta$) and then, after denoting $g = \nabla_\theta L$, we solve for $\eta$:

$$L(\theta) - \eta g^\top v \stackrel{!}{=} L^{\min} \qquad \Longrightarrow \qquad \eta = \frac{L(\theta) - L^{\min}}{g^\top v}. \tag{2}$$

First of all, note the clear separation between $g$, the estimator of the gradient of $L$ and $v$, the proposed update step. Moreover, it is easily seen that the final update step $\eta v$ is independent of the magnitude of $v$. In other words, the adaptation method only takes into account the "direction" of the proposed update. This decomposition into the gradient estimate and the update direction is the core principle behind the method and is also vital for its performance.

The update rule is illustrated in Fig. 1 for a quadratic (or other convex) loss. Here, we see (deceptively) that the proposed stepsize is, in fact, still conservative. However, in the multidimensional case, the

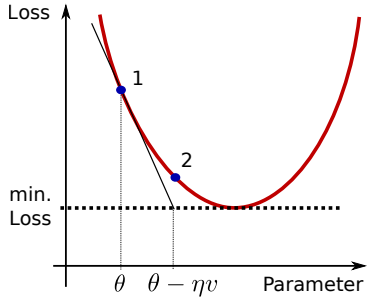

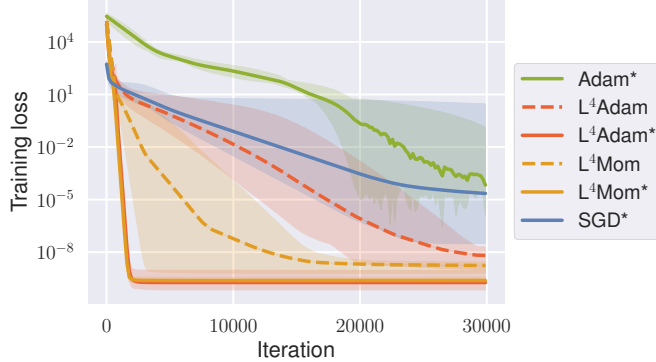

Figure 1: Illustration of stepsize calculation for one parameter. Given a minimum loss, the stepsize is such that the linearized loss would be minimal after one step. In practice a fraction of that stepsize is used, see Sec. 2.4.

Figure 2: Training performance on badly conditioned regression task with $\kappa(A) = 10^{10}$. The mean (in log-space) training loss over $5$ restarts is shown. The areas between minimal and maximal loss (after log-space smoothing) are shaded. For all algorithms the best stepsize was selected ($4 \cdot 10^{-5}$ and $10^{-3}$ for SGD and Adam respectively, and $\alpha = 0.25$ for both $L^4$ optimizers), except for the default setting without the "*". Note the logarithmic scale of the loss.

minimum will not necessarily lie on the line given by the gradient. That is why in real-world problems, this stepsize is far too aggressive and prone to divergence. In addition there are the following reasons to be more conservative: the problems in deep learning are (often strongly) non-convex, and actually minimizing the currently seen batch loss is very likely to not generalize to the whole dataset.

For these reasons, we introduce a hyperparameter $\alpha$ which captures the fixed fraction of the stepsize (2) we take at each step. Then the update rule becomes:

$$\Delta\theta = -\eta v = -\alpha \frac{L(\theta) - L^{\min}}{g^\top v} v, \tag{3}$$

Even though a few more hyperparameters will appear later as stability measures and regularizers, $\alpha$ is the main hyperparameter to consider. We observed in experiments that the relevant range is consistently $\alpha \in (0.10, 0.30)$. In comparison, for SGD the range of stable learning rates varies over multiple orders of magnitude. We chose the slightly conservative value $\alpha = 0.15$ as a default setting. We report its performance on all the tasks in Section 3.

## 2.3 Invariance to affine transforms of the loss

Here, we offer a partial explanation why the value of $\alpha$ stays in the same small relevant range even for very different problems and datasets. Interestingly, the new update equation (3) is invariant to affine loss transformations of the type:

$$L' = aL + b \tag{4}$$

with $a, b > 0$. Let us briefly verify this. The gradient of $L'$ will be $g' = ag$ and we will assume that the underlying optimizer will offer the same update direction $v$ in both cases (we have already established that its magnitude does not matter). Then we can simply write

$$-\alpha \frac{L'(\theta) - L^{\min'}}{g'^\top v'} v' = -\alpha \frac{aL(\theta) + b - aL^{\min} - b}{ag^\top v} v = -\alpha \frac{L(\theta) - L^{\min}}{g^\top v} v$$

and we see that the updates are the same in both cases. On top of being a good sanity check for any loss-based method, we additionally believe that it simplifies problem-to-problem adaptation (also in terms of hyperparameters).

It should be noted though that we lose this precise invariance once we introduce some heuristical and regularization steps in the following paragraphs.

### 2.4 Stability measures and heuristics

$L^{\text{min}}$ **adaptation:** We still owe an explanation of how $L^{\text{min}}$ is maintained during training. We base its value on the minimal loss seen so far. Naturally, some mini-batches will have a lower loss and will be used as a reference for the others. By itself, this comes with some disadvantages. In case of small variance across batches, this $L^{\text{min}}$ estimate would be very pessimistic. Also, the "new best" mini-batches would have zero stepsize.

Therefore, we introduce a factor $\gamma$ which captures the fraction of the lowest seen loss that is still believed to be achievable. Similarly, to correct for possibly strong effects of a few outlier batches, we let $L^{\text{min}}$ slowly increase with a timescale $\tau$. This reactiveness of $L^{\text{min}}$ slightly shifts its interpretation from "globally minimum loss" to "minimum currently achievable loss". This reflects on the fact that in practical settings, it is unrealistic to aim for the global minimum in each update. All in all, when a new value $L$ of the loss comes, we set

$$L^{\text{min}} \leftarrow \min(L^{\text{min}}, L),$$

then we use $\gamma L^{\text{min}}$ for the gradient update and apply the "forgetting"

$$L^{\text{min}} \leftarrow (1 + 1/\tau)L^{\text{min}}. \tag{5}$$

The value of $L^{\text{min}}$ gets initialized by a fixed fraction of the first seen loss $L$, that is $L^{\text{min}} \leftarrow \gamma_0 L$. We set $\gamma = 0.9$, $\tau = 1000$, and $\gamma_0 = 0.75$ as default settings and we use these values in all our experiments. Even though, we can not exclude that tuning these values could lead to enhanced performance, we have not observed such effects and we do not feel the necessity to modify these values.

**Numerical stability:** Another unresolved issue is the division by an inner product in Eq. (3). Our solution to potential numerical instabilities are two-fold. First, we require compatibility of $g$ and $v$ in the sense that the angle between the vectors does not exceed $90°$. In other words, we insist on $g^\top v \geq 0$. For L$^4$Adam and L$^4$Mom this is the case, see Section 2.5, Eq. (7). Second, we add a tiny $\varepsilon$ as a regularizer to the denominator. The final form of update rule then is:

$$\Delta\theta = -\alpha \frac{L(\theta) - \gamma L^{\text{min}}}{g^\top v + \epsilon}\, v\,, \tag{6}$$

with the default value $\varepsilon = 10^{-12}$.

### 2.5 Putting it together: L$^4$Mom and L$^4$Adam

The algorithm is called **L**inearized **L**oss-based optima**L** **L**earning-rate (L$^4$) and it works on top of provided gradient estimator (producing $g$) and an update direction algorithm (producing $v$), see Algorithm 1 in the Supplementary for the pseudocode. For compactness of presentation, we introduce a notation for exponential moving averages as $\langle\cdot\rangle_\tau$ with timescale $\tau$ using bias correction just as in [9] (see Algorithm 2 in the Supplementary).

In this paper, we introduce two variants of L$^4$ leading to two optimizers: (1) with momentum gradient descent, denoted by L$^4$Mom, and (2) with Adam [9], denoted by L$^4$Adam. We choose the update directions for L$^4$Mom and L$^4$Adam, respectively as

$$v = V_{Mom}(L, \theta) = \langle\nabla_\theta L(\theta)\rangle_{\tau_m} \qquad v = V_{Adam}(L, \theta) = \frac{\langle\nabla_\theta L(\theta)\rangle_{\tau_m}}{\sqrt{\langle|\nabla_\theta|^2 L(\theta)\rangle_{\tau_s}}}, \tag{7}$$

with $\tau_m = 10$ and $\tau_s = 1000$ being the timescales for momentum and (in case of L$^4$Adam) second moment averaging. In both cases, the choice of $g = G(L, \theta) = V_{Mom}(L, \theta)$ ensures $g^\top v \geq 0$, as mentioned in Section 2.4. Additional reasoning is that the averaged local gradient is in practice often a more accurate estimator of the gradient on the global loss.

## 3 Results

We evaluate the proposed method on five different setups, spanning over different architectures, datasets, and loss functions. We compare to the *de facto* standard methods: stochastic gradient descent (SGD), momentum SGD (Mom), and Adam [9].

For each of the methods, the performance is evaluated for the *best* setting of the stepsize/learning rate parameter (found via a fine grid search with multiple restarts). All other parameters are as follows: for momentum SGD we used a timescale of 10 steps ($\beta = 0.9$); for Adam: $\beta_1 = 0.9, \beta_2 = 0.999$, and $\varepsilon = 10^{-4}$. The (non-default) value of $\varepsilon$ was selected in accordance with TensorFlow documentation to decrease the instability of Adam.

In all experiments, the performance of the standard methods heavily depends on the stepsize parameter. However, in case of the proposed method, the *default* setting showed remarkable consistency. Across the experiments, it outperforms even the best constant learning rates for the respective gradient-based update rules. In addition, the performance of these default settings is also comparable with handcrafted optimization policies on more complicated architectures. We consider this to be the main strength of the $L^4$ method.

We present results for $L^4$Mom and $L^4$Adam, see Tab. 3 for an overview. In all experiments we strictly followed the **out-of-the-box** policy. We simply cloned an official repository, changed the optimizer, and left everything else intact. Also, throughout the experiments we have observed neither any runtime increase nor additional memory requirements arising from the adaptation.

As a general nomenclature, a method is marked with a $*$ if optimized stepsize was used. Otherwise (in case of $L^4$ optimizers), the default settings are in place. The Fashion MNIST [24] experiment can be found in the supplementary material as well as additional experiments with varying batch sizes as hinted in Tab. 3 by values in brackets.

**Running time:** Neither of the $L^4$ optimizers slows down network training in practical settings. By inspection of Equations (6) and (7), we can see that the only additional computation (compared to Adam or momentum SGD) is calculating the inner product $g^\top v$. This introduces two additional operations per weight (multiplication and addition). In any realistic scenario, these have negligible runtimes when compared to matrix multiplications (convolutions), which are required both in forward and backward pass.

## 3.1 Badly conditioned regression

The first task we investigate is a linear regression with badly conditioned input/output relationship. It has recently been brought into the spotlight by Ali Rahimi in his NIPS 2017 talk, see [18], as an example of a problem "resistant" to standard stochastic gradient optimization methods. For our experiments, we used the corresponding code by Ben Recht [17].

The network has two weight matrices $W_1, W_2$ and the loss function is given by

$$L(W_1, W_2) = \mathbb{E}_{x \sim \mathcal{N}(0,I)} \|W_1 W_2 x - y\|^2 \quad \text{s.t.} \qquad y = Ax \qquad (8)$$

where $A$ is a badly conditioned matrix, i. e. $\kappa(A) = \sigma_{\max}/\sigma_{\min} \gg 1$, with $\sigma_{\max}$ and $\sigma_{\min}$ are the largest and the smallest singular values of $A$, respectively. Note that this is in disguise a (realizable) matrix factorization problem: $L = \|W_1 W_2 - A\|_F^2$. Also, it is not a stochastic optimization problem but a deterministic one.

Figure 2 shows the results for $x \in \mathbb{R}^6, W_1 \in \mathbb{R}^{10 \times 6}, W_2 \in \mathbb{R}^{6 \times 6}, y \in \mathbb{R}^{10}$ (the default configuration of [17]) and condition number $\kappa(A) = 10^{10}$. The statistics is given for 5 independent runs (with randomly generated matrices $A$) and a fixed dataset of 1000 samples. We can confirm that standard

Table 1: **Overview of experiments.** The experiments span over classical datasets, traditional and modern architectures, as well as different batch sizes. The tested learning rates are denoted by $\alpha$ and marked with $*$ if chosen optimally via grid search. The optimal learning rates for the baselines vary while $L^4$ optimizers can keep a fixed setting and still outperforming in terms of training and test loss.

| Dataset | Arch | Batch size | $\alpha^*_{Mom/SGD}$ | $\alpha^*_{Adam}$ | $\alpha_{L^4Adam}$ | $\alpha_{L^4Mom}$ |
|---|---|---|---|---|---|---|
| Synthetic | 2-Layer MLP | - | | 0.0005 | 0.001 | **0.15** [0.25] | **0.15** [0.25] |
| MNIST | 3-Layer MLP | 64 [8,16,32] | 0.05 | 0.001 | **0.15** [0.25] | **0.15** [0.25] |
| CIFAR-10 | ResNet | 128 | 0.004 | 0.0002 | **0.15** | **0.15** |
| DNC | Recurrent | 16 [8, 32, 64] | 1.2 | 0.01 | **0.15** | **0.15** |
| Fashion MNIST | ConvNet | 100 | 0.01 | 0.0003 | **0.15** | **0.15** |

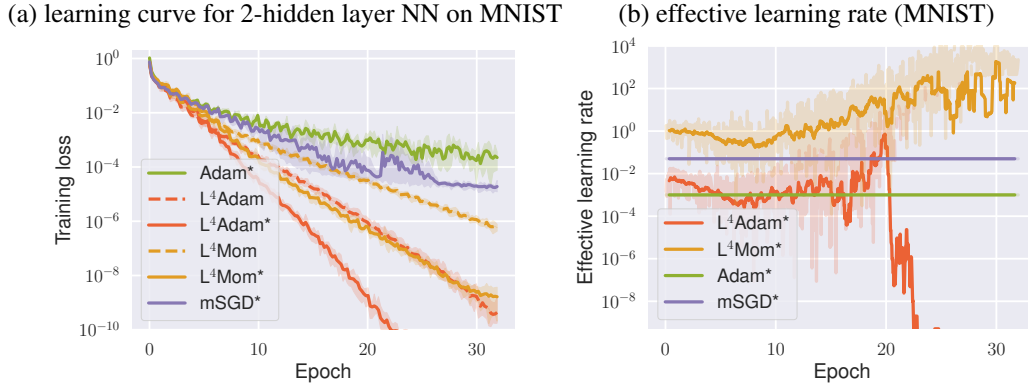

Figure 3: Training progress of multilayer neural networks on MNIST, see Section 3.2 for details. (a) Average (in log-space) training loss with respect to five restarts with a shaded area between minimum and maximum loss (after log-space smoothing). (b) Effective learning rates $\eta$ for a single run. The bold curves are averages taken in log-space.

optimizers indeed have great difficulty reaching convergence. Only a fine grid search discovered settings behaving reasonably well (divergence or too early plateaus are very common). The proposed stepsize adaptation method apparently overcomes this issue (see Fig. 2).

## 3.2 MNIST digit recognition

The second task is a classical multilayer neural network trained for digit recognition using the MNIST [11] dataset. We use the standard architecture with two layers containing 300 and 100 hidden units and ReLu activations functions followed by a logistic regression output layer for the 10 digit classes. Batch size in use is 64.

Figure 3 shows the learning curves and the effective learning rates. The effective learning rate is given by $\eta$ in (3). Note how after 22 epochs the effective learning of $L^4$Adam becomes very small and actually becomes 0 around 30 epochs. This is simply because by then the loss is 0 (within machine precision) on every batch and thus $\eta = 0$; a global optimum was found. The very high learning rates that precede can be attributed to a "plateau" character of the obtained minimum. The gradients are so small in magnitude that very high stepsize is necessary to make any progress. This is, perhaps, unexpected since in optimization theory convergence is typically linked to decrease in the learning rate, rather than increase.

Generally, we see that the effective learning rate shows highly nontrivial behavior. We can observe sharp increases as well as sharp decreases. Also, even in short time period it fully spans 2 or more orders of magnitude as highlighted by the shaded area, see Fig. 3(b). None of this causes instabilities in the training itself.

Even though the ability to generalize and compatibility with various regularization methods are not our main focus in this work, we still report in Tab. 2 the development of test accuracy during the training. We see that the test performance of all optimizers is comparable. This does not come as a surprise as the used architecture has no regularization. Also, it can be seen that the $L^4$ optimizers reach near-final accuracies faster, already after around 10 epochs.

**Comparison to other work:**  A list of papers reporting improved performance over SGD on MNIST is long (examples include [19, 13, 1, 2, 14]). Unfortunately, there are no widely recognized benchmarks to use for comparison. There is a lot of variety in choosing the baseline optimizer (often only the default setting for SGD) and in the number of training steps reported (often fewer than one epoch). In this situation, it is difficult to make any substantiated claims. However, to our knowledge, previous work does not achieve such rapid convergence as can be seen in Fig. 3.

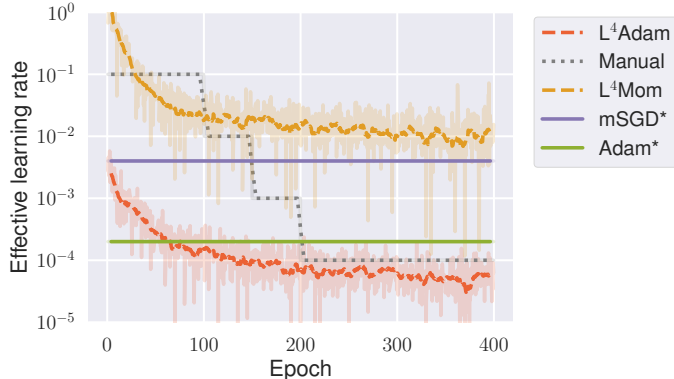

Figure 4: Effective learning rates $\eta$ for CIFAR10. The adaptive stepsize of L$^4$Mom matches roughly the hand-coded decay schedule (grey line) until 150 epochs. Both use the same gradient type.

### 3.3 ResNets for CIFAR-10 image classification

In the next two tasks, we target finely tuned publicly available implementations of well-known architectures and compare their performance to our default setting. We begin with the deep residual network architecture for CIFAR-10 [10] taken from the official TensorFlow repository [21]. Deep residual networks [8], or ResNets for short, provided the breakthrough idea of identity mappings in order to enable training of very deep convolutional neural networks. The provided architecture has 32 layers and uses batch normalization for batches of size 128. The loss is given by cross-entropy with $L^2$ regularization.

The deployed optimization policy is momentum gradient with manually crafted piece-wise constant stepsize adaptation. We simply replace it with default settings of L$^4$Mom and L$^4$Adam.

The first surprise comes when we look at Fig. 4, which compares the effective learning rates. Clearly, the adaptive learning rates are much more conservative in behavior compared to MNIST, possibly signaling for different nature of the datasets. Also the L$^4$Mom learning rate approximately matches the manually designed schedule (also for momentum gradient) during the decisive first 150 epochs.

Comparing performance against optimized constant learning rates is favorable for L$^4$ optimizers both in terms of loss and test accuracy (see Fig. 5). Note also that the two L$^4$ optimizers perform almost indistinguishably. However, competing with the default policy has another surprising outcome. While the default policy is inferior in loss minimization (more strongly at the beginning than at the end), in terms of test accuracy it eventually dominates. By careful inspection of Fig. 5, we see the decisive gain happens right after the first drop in the hardcoded learning rate. This, in itself, is very intriguing since both default and L$^4$Mom use the same type of gradients of similar magnitudes. Also, it explains the original authors' choice of a piece-wise constant learning rate schedule.

To our knowledge, there is no satisfying answer to why piece-wise constant learning rates lead to good generalization. Yet, practitioners use them frequently, perhaps precisely for this reason.

Table 2: Test accuracy after a certain number of epochs of (unregularized) MNIST training. The results are reported over 5 restarts.

| | Test accuracy in % | | | | | |
|---|---|---|---|---|---|---|
| | Adam | mSGD | L$^4$Adam | L$^4$Adam* | L$^4$Mom | L$^4$Mom* |
| 1 epoch | $95.7 \pm 0.3$ | $96.4 \pm 0.5$ | $95.9 \pm 0.7$ | $96.8 \pm 0.2$ | $95.4 \pm 0.5$ | $96.3 \pm 0.4$ |
| 10 epochs | $97.9 \pm 0.3$ | $98.0 \pm 0.1$ | $98.4 \pm 0.1$ | $98.3 \pm 0.0$ | $98.3 \pm 0.1$ | $98.3 \pm 0.1$ |
| 30 epochs | $98.0 \pm 0.4$ | $98.5 \pm 0.1$ | $98.4 \pm 0.1$ | $98.3 \pm 0.1$ | $98.4 \pm 0.1$ | $98.4 \pm 0.1$ |

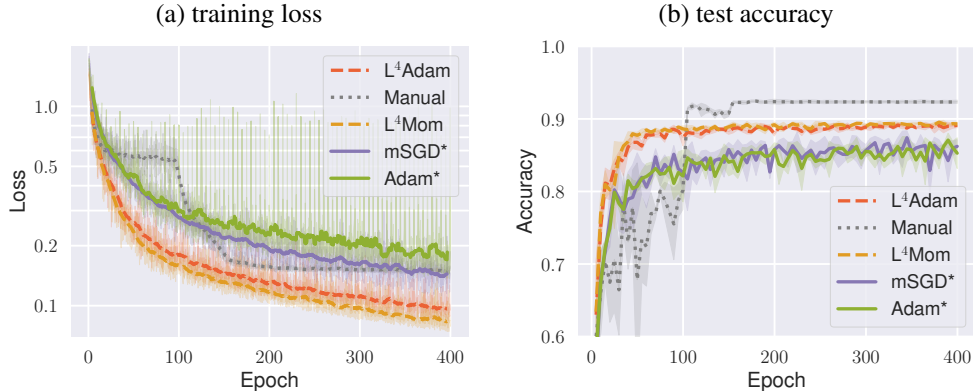

Figure 5: Training and test performance on ResNet architecture for CIFAR-10. Mean loss and accuracy are shown with respect to three restarts. The default settings of the $L^4$ optimizers perform better in loss minimization, however, become inferior in test accuracy after the first drop in learning rate of the baseline's learning rate schedule (see also Fig. 4). For Adam and mSGD, the best performing constant stepsizes $2 \cdot 10^{-4}$ and 0.004 were evaulated.

## 3.4 Differential neural computer

As the last task, we chose a somewhat exotic one; a recurrent architecture of Google Deepmind's Differential Neural Computer (DNC) [7]. Again, we compare with the performance from the official repository [4]. The DNC is a culmination of a line of work developing LSTM-like architectures with a differentiable memory management, e. g. [6, 20], and is in itself very complex. The targeted tasks have typically very structured flavor (e. g. shortest path, question answering).

The task implemented in [4] is to learn a REPEAT-COPY algorithm. In a nutshell, the input specifies a sequence of bits $a_n$ and a number of repeats $k$ while the expected output is a sequence $b_n$ consisting of $k$ repeats of $a_n$. The loss function is a negative log-probability of outputting the correct sequence.

Since, the ground truth is a known algorithm, the training data can be generated on the fly, and **there is no separate test regime**. This time, the optimizer in place is RMSProp [22] with gradient clipping. We found out, however, that the constant learning rate $10^{-3}$ provided in [4] can be further tuned and we compare our results against the improved value 0.005. We also used the best performing constant learning rates 0.01 for Adam and 1.2 for momentum SGD (both with the suggested gradient clipping) as baselines. The $L^4$ optimizers did not use gradient clipping.

Again, we can see in Fig. 6 that $L^4$Adam and $L^4$Mom performed almost the same on average, even though $L^4$Mom was more prone to instabilities as can be seen from the volume of the orange-shaded regions. More importantly, they both performed better or on par with the optimized baselines.

We end this experimental section with a short discussion of Fig. 6(b), since it illustrates multiple features of the adaptation all at once. In this figure, we compare the effective learning rates of $L^4$ and plain Adam. We immediately notice the dramatic evolution of the $L^4$ learning rate, jumping across multiple orders of magnitude, until finally settling around $10^3$. This behavior, however, results in a much more stable optimization process (see again Fig. 6), unlike in the case of plain Adam optimizer (note the volume of the green-shaded regions).

The intuitive explanation is two-fold. For one, the high gradients only need a small learning rate to make the expected progress. This lowers the danger of divergence and, in this sense, it plays the role of gradient clipping. And second, plateau regions with small gradients will force very high learning rates in order to leave them. This beneficial rapid adaptation is due to almost independent stepsize computation for every batch. Only $L^{\min}$ and possibly (depending on the underlying gradient methods) some gradient history is reused. This is a fundamental difference to methods that at each step make a small update to the previous learning rate. This is in agreement with [23], where the phenomenon was discussed in more depth.

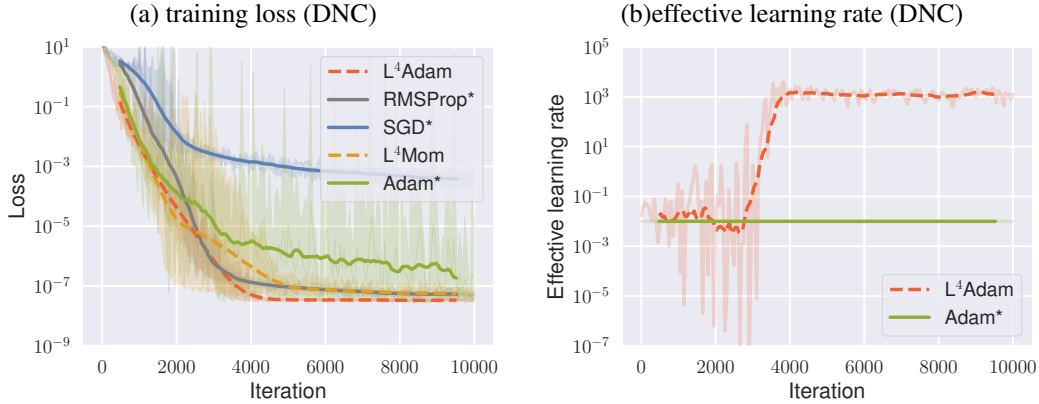

Figure 6: Training progress of the DNC. (a) Training loss (equals test loss) on the Differential Neural Computer architecture. See Fig. 2 for details. The $L^4$ optimizers use default settings, whereas RMSProp and Adam use best performing learning rates 0.005 and 0.01, respectively. We see high stochasticity in training, particularly with Adam. Both $L^4$ optimizers match or beat RMSProp in performance. (b) Effective learning rate $\eta$ of $L^4$Adam and plain Adam. The $L^4$Adam displays a huge variance in the selected stepsize. This however has a stabilizing effect on the training progress.

## 4 Discussion

We propose a stepsize adaptation scheme $L^4$ compatible with currently most prominent gradient methods. Two arising optimizers were tested on a multitude of datasets, spanning across different batch sizes, loss functions and network structures. The results validate the stepsize adaptation in itself, as the adaptive optimizers consistently outperform their non-adaptive counterparts, even when the adaptive optimizers use the default setting and the non-adaptive ones were finely tuned. This default setting also performs well when compared to hand-tuned optimization policies from official repositories of modern high-performing architectures. Although we cannot give guarantees, this is a promising step towards practical "no-tuning-necessary" stochastic optimization.

The core design feature, ability to change stepsize dramatically from batch to batch, while occasionally reaching extremely high stepsizes, was also validated. This idea does not seem widespread in the community and we would like to inspire further work.

The ability of the proposed method to actually drive loss to convergence creates an opportunity to better evaluate regularization strategies and develop new ones. This can potentially convert the superiority in training to enhanced test performance as discussed in the Fashion MNIST experiments.

Finally, Ali Rahimi and Benjamin Recht suggested in their NIPS 2017 talk (and the corresponding blog post) [17, 18] that the failure to drive loss to zero within machine precision might be an actual bottleneck of deep learning (using exactly the ill-conditioned regression task). We show on this example and on MNIST that our method can break this "optimization floor".

## 5 Acknowledgement

We would like to thank Alex Kolesnikov, Friedrich Solowjow, and Anna Levina for helping to improve the manuscript.

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
