[Supplementary Material · L4nips_2018_supplementary.pdf]

# L4: Practical loss-based stepsize adaptation for deep learning - Supplementary material

**Michal Rolínek and Georg Martius**
Max-Planck-Institute for Intelligent Systems Tübingen, Germany
michal.rolinek@tuebingen.mpg.de and georg.martius@tuebingen.mpg.de

## Abstract

In this supplementary material, we provide the missing details of the proposed stepsize adaptation and we report the results from additional experiments.

## A   Precise algorithmic description

The pseudocode of the $L^4$ algorithm is provided in Algorithm 1 and for moving averages with bias correction in Algorithm 2.

---

**Algorithm 1** $L^4$, a meta algorithm for stochastic optimization, compatible e. g. with momentum SGD, Adam. The default hyperparameters are: $\alpha = 0.15$, $\gamma = 0.9$, $\gamma_0 = 0.75$, $\tau = 1000$, and $\epsilon = 10^{-12}$.

---
**Require:** $\alpha$: Stepsize/fraction
**Require:** $\gamma, \gamma_0$: optimism loss improvement fraction
**Require:** $\tau$: timescale of forgetting minimum loss
**Require:** $L_t$: non-negative stochastic loss at time step $t$.
**Require:** $\theta_0$: initial parameter vector
**Require:** $V(L, \theta), G(L, \theta)$: gradient direction function, gradient step function
$t \leftarrow 0, \quad L^{\min} \leftarrow \gamma_0 \cdot L_0 \quad$ (fraction of initial loss)
**while** $\theta_t$ not converged **do**
   $t \leftarrow t + 1$
   $v \leftarrow V(L_t, \theta_{t-1}) \quad$ (gradient step)
   $g \leftarrow G(L_t, \theta_{t-1}) \quad$ (gradient estimator)
   $L_t^{\min} \leftarrow \min(L_{t-1}^{\min}, L_t) \quad$ (minimum loss)
   $\theta_t = \theta_{t-1} - \alpha \cdot \frac{L_t - \gamma \cdot L_t^{\min}}{g^\top \cdot v + \epsilon} \, v \quad$ (parameter update)
   $L_t^{\min} \leftarrow (1 + 1/\tau) \cdot L_t^{\min} \quad$ (forgetting minimum loss)
**end while**

---

**Algorithm 2** Bias corrected moving average

---
**Require:** $\tau$: timescape
$m_t \leftarrow 0 \quad$ (initialize mean with zero vector)
$t \leftarrow 0 \quad$ (initialize step counter)
**update_average**$(x)$:     ($x$: input vector to be averaged)
   $t \leftarrow t + 1$
   $m_t \leftarrow (1 - 1/\tau) \cdot m_{t-1} + 1/\tau \cdot x \quad$ (update average)
   **return:** $m_t / (1 - (1 - 1/\tau)^t) \quad$ (correct bias)

---

Table A2: Test accuracies on Fashion MNIST. Both L$^4$Mom and L$^4$Adam reach (slightly) higher accuracies. This effect is strenghtened by increasing the dropout rate to $p = 0.7$. Reported are mean and standard deviation of five independent restarts.

| L$^4$Adam ($p{=}0.7$) | L$^4$Mom ($p{=}0.7$) | L$^4$Adam | L$^4$Mom | Adam* | Mom* |
|---|---|---|---|---|---|
| $93.6 \pm 0.25$ | $93.45 \pm 0.15$ | $93.35 \pm 0.15$ | $93.25 \pm 0.2$ | $93.1 \pm 0.2$ | $93.0 \pm 0.15$ |

# B  Additional experiments

## B.1  Comparison with LMA

As an extension of the experiments on badly conditioned regression, we also include a comparison with the classical Levenberg-Marquardt algorithm (LMA) [1] which can be viewed as a Gauss-Newton method with a trust region. In Tab. A1 the speed of the algorithms, both in terms of the number of iterations as well as wall-clock time[1] is reported. The same comparison is also performed on an instance of twice the size (all dimensions doubled).

The results show that the gradients provided by LMA reach convergence in a much smaller number of steps. However, at the same time, LMA is significantly more computationally expensive since each step involves solving a least squares problem. This can be clearly seen from comparing performance on the problem sizes in Tab. A1.

Table A1: Comparison with Levenberg-Marquardt algorithm. Time and the number of gradient updates needed to reach convergence ($L < 10^{-8}$) is reported. The average is with respect to 5 restarts. Two problem setups are considered, the default from [2] ($x \in \mathbb{R}^6$, $W_1 \in \mathbb{R}^{10\times6}$, $W_2 \in \mathbb{R}^{6\times6}$, $y \in \mathbb{R}^{10}, \kappa(A) = 10^{10}$) and its "scaled up by two" version. Stepsize $\alpha = 0.3$ was selected for LMA as the best performing one, and $\alpha = 0.25$ is chosen for both L$^4$ optimizers.

| Method | Steps | Time (s) |
|---|---|---|
| 96 trainable weights | | |
| L$^4$Adam* | $2325 \pm 765$ | $1.4 \pm 0.5$ |
| L$^4$Mom* | $1606 \pm 32$ | $1.0 \pm 0.02$ |
| LMA* | $68 \pm 3$ | $3.0 \pm 1.4$ |
| 192 trainable weights | | |
| L$^4$Adam* | $2222 \pm 311$ | $2.1 \pm 0.4$ |
| L$^4$Mom* | $1933 \pm 116$ | $1.8 \pm 0.2$ |
| LMA* | $212 \pm 114$ | $123 \pm 64$ |

## B.2  Fashion MNIST

The Fashion MNIST dataset [5] is a drop-in replacement for MNIST that is considered to better represent modern computer vision tasks. We ran it on a TensorFlow official implementation of a ConvNet for MNIST [4]. The architecture consists of two convolutional layers followed by two fully connected layers that a have a dropout [3] in between. By default, the batch size is 100 and the optimizer is Adam.

We see in Fig. A1, that both L$^4$ optimizers work out-of-the-box and despite the presence of dropout during training, both achieve a loss that is roughly an order of maginute lower than the losses of optimized baselines. This leads to a mild gain in test accuracy as can be seen in Tab. A2.

Such low training loss of L$^4$ optimizers despite the presence of dropout suggests increasing the dropout rate in hope for better generalization. And indeed when switching from the default rate $p = 0.4$ to value $p = 0.7$, one can see (also in Tab. A2) an additional gain in test accuracy.

Figure A1: Training performance on Fashion MNIST. Default $L^4$ optimizers reach lower level of the loss depsite the presence of dropout (rate $p = 0.4$).The optimized learning rates for mSGD and Adam were 0.01 and 0.0003, respectively. Results are reported over five independent restarts.

(a) $L^4$Adam

(b) $L^4$Mom

Figure A2: Sweeping over batch sizes for MNIST. The plotted curves are averages over five restarts (with smoothing in log-spcae). All smaller batch sizes outperform the batch size 64 used in the main paper.

Although generalization performance was not our main focus in this paper, we firmly believe that superior performance in optimization is convertible to better results in test time. This case of increasing the dropout rate is one promising example of it.

## B.3    Sweeping over batch sizes

Since $L^4$ recomputes the stepsize individually for each batch, it is natural to investigate how its performance depends on the batch size. For this experiment we chose the MNIST and DNC datasets, since there $L^4$ displayed the most variance in the effective learning rates, and thus behaved "most different" from standard optimizers. Of particular interest is the "high variance" regime of small batch sizes, where the stepsize adaptation can be expected to evolve the learning rate rapidly. The same default setting ($\alpha = 0.15$) of both $L^4$Mom and $L^4$Adam was consistently applied.

In both cases we swept over batch size 8, 16, 32, and 64. In the experiments from the main text of the paper, the selected batch sizes were 64, and 16, respectively for MNIST and DNC.

The results for MNIST are plotted in Fig. A2. It turns out that the original setting is the least favorable for both $L^4$ optimizers. In fact, the performance increases with decreasing the batch size.

For DNC, we report the performance in Fig. A3. We also observe that $L^4$ favors small batch sizes. Here batch size 8 is probably the limit of what $L^4$ can tolerate. This is not directly visible on the loss

(a) $L^4$Adam

(b) $L^4$Mom

Figure A3: Sweeping over batch sizes for DNC. Results are averaged over five independent restarts. The $x$-axis is normalized per number of data points processed. In case of $L^4$Mom (b) with batch size 8, one run diverged (after already having reached convergence). The reported curve is the average over the remaining four runs.

curves but in fact one of the runs of $L^4$Mom diverged (after processing 50000 examples and reaching $10^{-7}$ loss). In all other cases, good level of convergence was reached.

In conclusion, adapting the stepsize for every batch shows to be gradually more beneficial as we lower the batch sizes (loss estimates increase in variance). This is a further validation for applying learning rates that are highly varying from mini-batch to mini-batch.

Although the batch sizes in both cases range almost over an order of magnitude, no severe deterioration of performance was ever detected.

## Footnotes

[1]The experiments were conducted on a machine with i7-7800X CPU @ 3.50GHz with 8 cores.