[Reviews · NeurIPS 2018]

Reviewer 1



The authors propose a rather simple method to accelerate training of neural networks by doing a form of linear extrapolation to find the optimal step size. The idea is simple enough, and the authors provide no theoretical analysis for this method. In a way, most optimization papers focus on convex methods, which they analyze well, and then they apply them to non-convex problems (e.g. neural networks). I’m not sure if the authors stumbled upon a very smart idea, or if they are a bit mistaken about how good their method is. I am not very confident in my ability to determine the validity of this method. It seems to me that this could be a good idea that other people could try, and we could see how well it performs on other deep learning problems. I would accept this paper, but I would not be surprised to find out that some expert in optimization would chime in and point out why this method is a bad idea and it has already been tried.

Reviewer 2



The paper proposes a scheme for adaptive choice of learning rate for stochastic gradients descent and its variants. The key idea is very simple and easy to implement: given the loss value L at the global minimum, L_min, the idea is to choose learning rate eta, such that the update along the gradient reaches L_min from the current point i.e. solving L(theta-eta*v)=L_min in eta, where v is for example dL/dtheta in gradient descent. Due to nonlinearity of L, the authors suggest linearizing L around theta: L(theta)+(-eta*v)^T dL/dtheta=L_min which yields eta=(L(theta)-L_min)/(v^T dL/dtheta). Finally, to make the adaptive learning rate pessimistic to the possible linearization error, the authors introduce a coefficient alpha, so the effective learning rate used by the optimizer is eta*alpha. The authors empirically show (on badly conditioned regression, MNIST, CIFAR-10, and neural computer) that using such adaptive scheme helps in two ways: 1. the optimization performance is less sensitive to the choice of the coefficient alpha vs the learning rate (in non-adaptive setting), and 2. the optimizer can reduce the loss faster or at worst in equal speed with commonly used optimizers. At the same time, the paper has some shortcomings as admitted by the authors: 1. There is no theoretical justification for the proposed scheme. 2. Despite training faster, the resulted model may generalize worse than models trained with non-adaptive step size. In spite of the above shortcomings, I think this is a nice paper which may motivate further research around the proposed idea and eventually create an impact. Some comments that may improve the manuscript: 1. I think providing a graphical illustration on the trajectories taken by this scheme vs plain gradient descent can be very insightful and maybe provide some intuition why the idea works. Perhaps this can be shown on a simple task: a poorly conditioned quadratic function. We know that in this case the gradient may not necessarily point toward the minimum. So it is interesting to visually inspect how by merely modifying the step size, and yet moving along the gradient directions over this poorly condition surface, the adaptive step size method can reach the minimum faster. 2. For CIFAR-10 graphs (Figure 5), what happens after 250 epochs? Is it that all methods eventually converge to the same loss value or some methods get stuck at larger values? One major concern I have about this paper is the way L_min is used. Is L_min defined locally (i.e. the minimal point along the gradient ray only) or globally (over the entire space)? I suppose it is global, because the notation does not show any dependency of L_min on theta. If it is defined globally, then how would you justify solving L(theta-eta*v)=L_min, because this equation may not even have a solution( and I think often it does not!). For example, for a poorly conditioned quadratic, the direction of gradient may not necessarily point toward the minimum. Hence, no matter how much you move along the gradient direction, you never reach the minimum value L_min. Therefore, It seems the idea of solving L(theta-eta*v)=L_min has some fundamental problem. POST REBUTTAL COMMENTS: I read author's response. I was not convinced about L_min justification. Specially, the fact that setting L_min to "0" did not work shows the crticality of L_min for the algorithm to work. However, I am also not aware of a cheap workaround, so I believe the setting is okay although not idea. I keep my rating as it was.

Reviewer 3



The paper proposed a new learning rate adaptation method. It approximates the currently lowest achievable loss, and derive the stepsize that achieve the loss under a linear approximation to the loss function. In general, the empirical results look great. Fantastic results are achieved across multiple benchmarks ranging from a very badly conditioned toy task to modern neural networks training. More importantly, almost no tuning is needed across these benchmarks. It is very exciting to see such method come out. However, the paper also has a major weakness. The adaptation rule proposed is exactly the same to Polyak's step length, with a slight change that the update direction in the original Polyak's step length is now changed from gradient direction to a general update direction. See a description of this in the section 4 of the course note: https://web.stanford.edu/class/ee364b/lectures/subgrad_method_notes.pdf. I think the paper should mention this fact and humbly admit it. Instead of rebranding this as a new algorithm, just admitting that this work revisits Polayk's update rule. That being said, it does not degrade the contribution the paper made to demonstrate the effectiveness of this algorithm, with extra design of how to estimate the minimal loss. Another comment I want to make is that although the results look fantastic, there is little discussion in what aspect the method can help improve. It will be great to see some discussion of understanding the adaptation method, at least in some toy task domain, by probing into the properties of the adaptive rule in some way. Lastly, I do want to see some wall clock time comparisons between your method and the baseline. Such standard comparisons should be done for a optimization method paper.